# Oncologic Drugs Approval in Europe for Solid Tumors: Overview of the Last 6 Years

**DOI:** 10.3390/cancers14040889

**Published:** 2022-02-11

**Authors:** Rosa Falcone, Pasquale Lombardi, Marco Filetti, Simona Duranti, Antonella Pietragalla, Alessandra Fabi, Domenica Lorusso, Valeria Altamura, Francesco Paroni Sterbini, Giovanni Scambia, Gennaro Daniele

**Affiliations:** 1Phase 1 Unit, Fondazione Policlinico Universitario A. Gemelli IRCCS, 00168 Rome, Italy; rosa.falcone@policlinicogemelli.it (R.F.); pasquale.lombardi@guest.policlinicogemelli.it (P.L.); marco.filetti@guest.policlinicogemelli.it (M.F.); valeria.altamura@policlinicogemelli.it (V.A.); francesco.paronisterbini@policlinicogemelli.it (F.P.S.); 2Scientific Directorate, Fondazione Policlinico Universitario A. Gemelli IRCCS, 00168 Rome, Italy; simona.duranti@policlinicogemelli.it (S.D.); antonella.pietragalla@policlinicogemelli.it (A.P.); alessandra.fabi@policlinicogemelli.it (A.F.); domenica.lorusso@policlinicogemelli.it (D.L.); giovanni.scambia@policlinicogemelli.it (G.S.); 3Unit of Precision Medicine in Breast Cancer, Scientific Directorate, Fondazione Policlinico Universitario A. Gemelli IRCCS, 00168 Rome, Italy; 4Department of Life Science and Public Health, Università Cattolica del Sacro Cuore, 00168 Rome, Italy

**Keywords:** oncologic drug, EMA approval, solid tumor, indication, primary end-point, quality of life

## Abstract

**Simple Summary:**

We present a systematic overview of EMA-approved oncologic drugs, both as new or extensions of indications, over 6 years, from 2015 to 2020. Our analysis confirms a long-standing trend in drug development for which most of the efforts led to broadening indications of pre-existing molecules, with immunotherapy and signal transduction inhibitors contending the leadership. Many drugs with the same or a very similar mechanism of action are approved for the same indication without a direct head-to-head comparison. Moreover, we show that the majority of drugs entered the market without evidence of overall survival or quality of life benefit but based on surrogate outcomes. Our data support some considerations about the modern drug development in oncology, where the goal is to provide the patients with the fastest possible access to new drugs. The debate on the optimal methodology to develop new drugs remains an unmet need.

**Abstract:**

(1) Background: Drug development in oncology is changing rapidly. The aim of the present study was to provide an insight into the features of anti-tumor drugs approved in Europe; (2) Methods: We included all the indications for solid tumors issued by the European Medicines Agency (EMA) between 2015 and 2020. We extracted data from European Public Assessments Reports (EPAR), including drug name, mechanism of action, setting, features of pivotal clinical trials, primary end-points, quality of life (QoL); (3) Results: In the explored period, EMA issued 132 new indications (81 indications’ extensions) for 62 oncology drugs. In about half of indications (47%), the approval was biomarker-based. Immune check point inhibitors (ICIs) and signal transduction inhibitors were the two most representative drug categories (62%). Most of the indications were for the advanced setting (91%) and front-line therapy (66%). The most common tumor types were non-small cell lung cancer (24%), breast (16%), and melanoma (10%). Two thirds of the indications (73%) were approved based on phase III trials. Overall survival (OS) represented the primary end-point only in 39% of indications, mainly limited to advanced setting (98%) and ICI trials (80%). Almost all (94%) cell cycle and DNA repair mechanism inhibitors were approved based on progression free survival (PFS) data. In pivotal trials with signal transduction inhibitors, objective response rate (ORR) was the prevalent (45%) primary end-point. QoL was never considered as primary end-point; (4) Conclusions: In this analysis, we intended to offer an updated picture of the recent drug development in oncology. Most of the efforts led to broadening indications of pre-existing molecules, with signal transduction inhibitor and ICIs contending the leadership. Twenty-seven percent of the indication were approved without a phase III trial. The majority of drugs entered the market without evidence of OS or QoL benefit but based on surrogate outcomes.

## 1. Introduction

Clinical trials represent the main source of evidence used by regulatory agencies for drug approvals. To implement a new indication, a positive benefit-risk balance must be demonstrated and based on evidence of quality, safety, and efficacy.

According to European Medicines Agency (EMA) guidelines [1], demonstrating an advantage in overall survival for a new indication is the most reliable outcome both from a clinical and methodological perspective. Other survival (progression free survival, PFS; disease free survival, DFS) outcomes are acceptable alternative end-points, often defined, erroneously, as surrogate end-points. Moreover, EMA recognizes the value of patient-reported outcome (PRO) measures from a regulatory point of view in the approval process of anti-cancer drugs [2].

PROs provide patients’ perception on the impact of disease and treatments on symptoms and quality of life [3]. The use of PRO outcomes as end-points in clinical trials has been encouraged, although the methodology is still lacking [4].

Over the last decades, new classes of anti-cancer medicines have been developed, and are under development. The challenge to quickly perform extensive molecular profiling and the emergence of an increasing number of selective inhibitors, immune-modulating agents, and other non-cytotoxic drugs required the change of clinical study design [5]. On one side, this fostered hope in shortening the time for access to new treatments for severe clinical conditions [6], but on the other side raised several methodological issues [7].

EMA’s Committee for Medicinal Products for Human Use (CHMP) may grant a conditional marketing authorization for medications that address unmet clinical needs, used to treat patients with life threatening or debilitating diseases. These drugs are approved based on less comprehensive data (smaller studies without comparisons, using surrogate end-points) than normally required. This strategy raised issues on the uncertainty of their clinical benefit because some drugs approved with first-analysis data turn out to be, over time, less effective than initially shown [8,9].

Here, we offer a systematic overview of EMA-approved oncologic drugs, both as new or extensions’ indications, over a 6-year period.

## 2. Materials and Methods

We searched new medicines for solid tumors, in adults, that received marketing authorization by the EMA between January 2015 and December 2020. We also included the extension of therapeutic indications of drugs originally approved for the treatments of cancer in adults (i.e., extension to adolescents and children). We excluded indications limited to pediatric settings, supportive therapy, generics, and biosimilars.

Data were obtained from the EMA website (www.ema.europa.eu, 16 June 2021), exploiting the European Public Assessments Reports (EPAR) that were manually consulted. EPARs are official regulatory documents that summarize the evidence used by the EMA to grant a new indication.

We collected data about positive recommendations on new medicines or extensions of indications of previously approved drugs: name of medicines, marketing authorization, date of approval, orphan drug designation, therapeutic indication, setting of disease (not advanced versus advanced/metastatic), line of therapy, and features of main study used for approval (phase, randomization, crossover, sample size, control arm, experimental arm, end-points, quality of life (QoL)).

A new indication was the first approval in Europe for a given medication. Extension of indication regarded medicines that are already available on the European market but their new indication for use (new population/combination/setting/tumor) has been approved. Two types of marketing authorizations were distinguished: the regular marketing authorization and the conditional one.

This latter may be granted if the medicine fulfils an unmet medical need, the benefit-risk ratio of the medicine is positive, additional data have to be provided in the post-marketing period. Orphan designation is granted for the development of medicines used to treat, diagnose, or prevent rare diseases, giving access to economic incentives and scientific support.

Oncology drugs were categorized in seven groups based on the mechanisms of action: (1) immune-check point inhibitors (ICIs) (anti PD-1/PD-L1/CTLA4); (2) signal transduction inhibitors, that include both monoclonal antibodies and kinase inhibitors blocking selective and aberrantly activated intracellular pathways (EGFR, ALK, HER2, BRAF, etc.); (3) hormone therapy; (4) chemotherapy; (5) drugs interfering with cell cycle and DNA repair mechanisms (i.e., PARP inhibitors); (6) angiogenesis inhibitors, both monoclonal antibodies and multi-kinase inhibitors; and (7) radiometabolic compounds.

Information on biomarker and subgroup analysis-based approvals were retrieved by EPAR. Biomarkers are biological markers of any type that are objectively measurable. In our case, they refer to molecular characteristics (driver genes, immunological markers, protein expressed by the tumor). The approval was defined as biomarker-based if the indication was limited to patients affected by solid tumors expressing a particular biomarker. An approval was defined based on subgroup analysis if granted for a subgroup of the initial population defined by the study.

Additional information about the starting data and publication of the pivotal trial were extracted from the ClinicalTrials.gov (accessed on 9 February 2022) website and PubMed, respectively.

Information on quality of life, when assessed and reported for EMA evaluation was categorized in three items: (1) no differences between control and experimental arm; (2) superiority of experimental arm; (3) superiority of control arm. Information was reported as not applicable if a control arm was absent or data were not reported.

Data were extracted independently by three authors (P.L., M.F., S.D.) and collected into a Microsoft Office Excel 2016 form. Information was double-checked by another co-author (R.F.). Disagreements in interpretation were solved by discussion among the authors.

Quantitative and qualitative variables were depicted with descriptive statistics. In order to explore a potential trend over the years in the Hazard Ratio (HR) for both OS and PFS, we set two linear regression models with year of approval as independent variable and HR for OS and PFS as dependent variable, respectively. Each model was adjusted for type of disease and setting (Advanced vs. Early). All the analyses were performed using STATA (STATA/IC 16.1; StataCorp).

## 3. Results

From 2015 to 2020, EMA issued 132 indications, with 61% (*n* = 81) indication extensions for 62 oncology drugs. In about half of indications (*n* = 62, 47%), the approval was biomarker-based. Mean time from starting date of the pivotal study to drug approval was 53.9 (range, 12–170) months.

### 3.1. Drug Class, Market Authorization, and Orphan Designation

Signal transduction inhibitors and ICIs were the two most representative drug categories, accounting for more than half of indications (*n* = 82, 62%). Details about class drug distribution are reported in Table 1.

Among ICIs, pembrolizumab, nivolumab, and atezolizumab received the vast majority of approvals: 13 (32.5%), 10 (25%), and 8 (20%), respectively. Olaparib, with six indications, was in fourth place in the list of most approved drugs (Appendix A).

EMA’s CHMP granted a regular market authorization for the vast majority of indications (*n* = 117/132, 89%). Fifteen (11%) received a conditional market authorization. Signal transduction inhibitors represent almost all (*n* = 13/15, 87%) of drugs approved with a conditional market authorization. Eight (*n* = 8/62, 13%) drugs were recognized as orphan medicine at the time of granting the marketing authorization and three of them were approved with a conditional market authorization.

Over time, half (*n* = 4/8) lost the designation of orphan drugs. Olaratumab, originally (in 2015) designated as orphan medicine and entering the European market with a conditional market authorization for the treatment of soft tissue sarcoma, in combination with doxorubicin, was later (in 2019) withdrawn because of the negative results reported by the phase III ANNOUNCE trial.

### 3.2. Approvals by Tumor and Setting

The vast majority of indications (*n* = 120, 91%) were for the advanced/metastatic setting and front-line therapy (*n* = 79, 66%). Indications were especially limited to some disease: 32 (24%) regarded non-small-cell lung cancer (NSCLC), 21 (16%) for breast, 13 (10%) for melanoma, and 10 (8%) for ovarian cancer (Table 2).

In NSCLC, more than half of approvals concerned signal transduction inhibitors (*n* = 18/32, 56%), followed by ICIs (*n* = 10/32, 31%). In melanoma, ICI and signal transduction inhibitors were confirmed as prevalent indications (62 and 38%, respectively).

In contrast, in ovarian cancer, drugs involved in cell cycle and DNA repair mechanisms reached 80% of approvals. For breast, the same percentage of approvals (43%) pertained to drugs involved in cell cycle/DNA repair mechanisms and signal transduction inhibitors.

### 3.3. Features of Main Approved Studies

Two thirds of the indications (*n* = 97/132, 73%) were approved based on phase III trials and 16% based on phase II trials. Twelve indications (9%) were based on early phase (I or I/II) trials (Table 3).

The approvals based on early phase studies regarded especially targeted therapy (anti ROS1, anti ALK, and anti-RET) in NSCLC and some rare disease (medullary and non-medullary thyroid cancer, soft tissue sarcoma, GIST). Moreover, agnostic marker-based approval of entrectinib for the treatment of adult and pediatric patients 12 years of age and older, with solid tumors expressing a neurotrophic tyrosine receptor kinase (NTRK) gene fusion fell within this definition.

Median sample size of main studies was 500 (interquartile range, 265–821 patients).

Most pivotal trials were randomized (*n* = 109/132, 82%), 89% (*n* = 97/109) of which were phase III trials. More than half (*n* = 13/21, 62%) of phase II trials were non-randomized.

Most (*n* = 100/132, 76%) indications were based on studies with a superiority hypothesis. About 20% had exploratory/descriptive or no a priori hypothesis. In the former case, randomized phase III study design almost always (*n* = 90/100, 90%) supported the indication. In the latter, solely early phase (I, I–II, and II) trials were numbered, most of which (*n* = 22/24, 92%) did not include randomization and mainly (*n* = 17/24, 71%) leading to signal transduction inhibitors’ approval. Interestingly, 7.6% of the indications were granted based on subgroup analysis.

In the experimental arm, there was a slight prevalence of monotherapy (*n* = 77/132, 58%) over treatment combination. The control arm was represented by an active treatment, with or without placebo, in two thirds of cases (65%). A comparison was lacking in 17% of indications. In particular, more than one third (*n* = 15/42, 36%) of signal transduction inhibitor indications was approved without a treatment comparison.

For a minority of indications (*n* = 15/132, 11%), crossover was allowed into the studies, which were mostly (*n* = 12/15, 80%) phase III trials where the experimental arm was compared with active treatment, with or without placebo.

Overall survival was the primary end-point in 42% (*n* = 51/120) of approved indications in the advanced setting compared to 8% in early stage disease (*n* = 1/12), where survival outcomes other than OS (interval disease free survival, metastases free survival, relapse free survival) prevailed (*n* = 9/12, 75%). For advanced-metastatic setting, OS and PFS were used as primary end-point similarly (42 and 35%, respectively).

In early phase (I, I–II, II) studies, radiological or pathological response rate was the main end-point (*n* = 26/33, 79%) whilst in more advanced phase trials 97% (n = 96/99) of indications were approved based on studies having survival outcomes as primary end-points. Co-primary end-points were present in one fourth of cases (*n* = 33/132, 25%). In most of these cases (*n* = 26/33, 79%), OS was a co-primary end-point.

Almost all (*n* = 17/18, 94%) cell cycle and DNA repair mechanism inhibitors were approved based on PFS data as main study end-points; OS was the preferred end-point in studies with immunotherapy (*n* = 32/40, 80%) and chemotherapy (*n* = 6/8, 75%). In pivotal trials with signal transduction inhibitors, the plethora of end-points is diverse with a prevalence of response rate (*n* = 19/42, 45%) overall.

Median OS across all indication was 12.30 (IQR, 10.35–16.57) months and the median HR was 0.71 (IQR, 0.63–0.77). The median overall survival increase between experimental and control arm was 2.81 (IRQ, 1.92–4.60) months. Median PFS was 11 (IQR, 7.65–16.60) months and median HR was 0.56 (IQR, 0.47–0.65). The median increase in PFS was 4.80 (IQR, 2.53–8.45) months.

In Appendix A, we report the mean HR for disease/setting and type of drug, supporting the approval by EMA. We did an exploratory evaluation of the effect size changing over time for accepted indications. Adjusting for the setting of approval, neither OS nor PFS variation of HR over-time was significant (R2 = 0.029 and R2 = 0.004, respectively).

Quality of life was evaluated in the main study for 81% (*n* = 107/132) of indications with some data available in EPAR for approval in 67% (*n* = 88/132) of them. If quality of life was reported, for most of indications (*n* = 55/88, 62.5%), no difference between the experimental and control arm was demonstrated. In one fourth of indications (*n* = 21/88, 24%), the experimental therapy showed better quality of life. Quality of life data, as expected, were increasingly reported in proportion to the advancement of phases of trials (78% for phase III trials, 45% for phase II trial, 33% for phase I-II trial). Interestingly, 7.6% of the indications were granted based on subgroup analysis.

## 4. Discussion

With this work, we aimed to provide a comprehensive snapshot of the drugs approved for cancer in the last 6 years in Europe. First, it should be noticed that, compared with the previous decades, there is an increasing number of new indications approved by EMA in 2015–2020. Apolone et al. [10], over a period of 10 years, from 1995 to 2004, identified 14 anticancer drugs for 27 different indications. Subsequently, from 2009 to 2013, 51 indications were approved [11]. Surprisingly, from 2015 to 2020 more than twice the previous number of approved medications (132) entered the European market.

The picture that can be drawn from these data shows that, many drugs, with a similar mechanism of action (“me-too”), concentrated in few tumor types, are mainly approved based on surrogate end-points. Interestingly, there is also a non-negligible use of conditional approval based on data from early studies. These data support some considerations about modern drug development in oncology, where the goal is to provide the patients with the fastest possible access to new drugs.

Many drugs with the same, or a very similar mechanism of action, are approved for the same indication without a direct head-to-head comparison. Among these, there are crizotinib, ceritinib, alectinib, lorlatinib, and osimertinib in actionable NSCLC, selpercatinib in RET fusion disease, entrectinib and larotrectinib in solid tumors expressing a neurotrophic tyrosine receptor kinase (NTRK) gene fusion, pembrolizumab in urothelial patients not eligible for cisplatin-containing chemotherapy, avelumab for Merkel cell carcinoma, and trastuzumab deruxtecan in HER2 positive breast cancer.

Recently, in a cross-sectional study, V. Prasad et al. [12], looking at all anticancer drugs approved by the FDA from January 2009 to December 2020, showed that approvals based on a new mechanism of action represented a minority (16%) of all approvals, compared to next-in-class drugs or extensions of previous indications to other cancers.

Interestingly, the vast majority of “me-too” drugs were approved based on a surrogate endpoint. Moreover, albeit in a minority of cases, the approval was based on a non-comparative/single arm trial.

Lung cancer, breast cancer, and melanoma, compared to the past, are confirmed as the three tumors with the highest number of new indications, with NSCLC leading the standings (24%) [13].

Signal transduction inhibitors and ICIs were the two prevalent drug categories that entered the market, accounting for 32% and 30%, respectively. Signal transduction inhibitor or targeted anti-cancer agents target specific molecules required for cell growth, tumorigenesis, and survival [14]. The large success of these drugs comes from their specificity, thus they are expected to have few side effects. Between 2015 and 2020, EMA approved 29 signal transduction inhibitors for 42 indications.

ICIs targeting the PD-1/PD-L1 axis are now approved, alone or in combination with other agents, in the treatment of a large number of malignancies (melanoma, NSCLC, head and neck squamous cell carcinoma, colorectal cancer, breast cancer, urothelial carcinoma, etc.) [15]. There are currently six PD-1/PD-L1 inhibitors EMA-approved in the treatment of solid tumors. Among all medications, from 2015 to 2020, pembrolizumab, nivolumab, and atezolizumab received the vast majority of approvals: 32.5%, 25%, and 20%. In a minority of indications, the use of these ICIs is bound to the presence of a predictive biomarker (PD-L1 expression by immunohistochemistry, microsatellite instability-high (MSI-H), or mismatch repair deficient (dMMR)).

Overall survival was the primary end-point for 39% of approved drugs, dropping to 8% in non-advanced disease. Other survival outcomes amounted to 39%. Pharmacokinetics, ORR, and pathological complete response (pCR) represented the remaining 22% of primary end-points. The use of primary end-points other than OS is still a matter of debate. Using surrogate end-point of OS is acceptable in few oncologic conditions. In advanced ovarian cancer, in 2017, the Gynecologic Cancer InterGroup recommended that PFS can be used as a primary end-point instead of OS [16]. A recent meta-analysis questioned this point and did not recognize PFS as a surrogate end-point for OS in randomized clinical trials of first-line treatments of advanced ovarian cancers [17]. The author suggested that if PFS is chosen as a primary end-point, OS must be measured as a secondary end-point and PFS must be supported by additional end points (i.e., PROs). In our analysis, two thirds (73%) of indications for ovarian cancer were based on PFS as primary outcome. Pathological complete response (pCR) has been demonstrated to be an acceptable end-point in the neoadjuvant setting for drugs used to treat patients with high-risk, early stage, HER2 positive breast cancer [18]. pCR was associated with long term outcomes (event-free survival and OS). It was used in NEOSPHERE trial for the approval of pertuzumab, in combination with trastuzumab, in neoadjuvant setting of HER2 positive high-risk breast cancer [19].

In NSCLC, OS should be selected as the primary end-points in confirmatory trials and for maintenance studies [20]. OS is also the primary end-point for immunotherapy trials [21], data confirmed also by our work, with OS representing the preferred end-point (80%).

The median increase in OS extrapolated by these data was low (2.8 months) and it did not change across the last six years and among drug classes. The magnitude of OS benefit was stable from 2009 to currently. Davis et al., reported a median value of 2.7 months in the explored period 2009–2013 [11]. Previous reports suggested that even in the post-marketing period, after 3–5 years from EMA approval, for about 30–50% of therapies, there was uncertain evidence showing a benefit in extended or improved life [22]. Similar findings are reported in the analysis published by Ladanie et al., about drug indications approved by the American Drug Agency, the Food and Drug Administration over a period of 17 years. At the time of drug approval, limited evidence of prolonged survival is available and they are associated with a small absolute increase in overall survival of 2.4 months [23].

Many authors question the utility of a fast-track approval procedure such as conditional marketing authorization because it may allow ineffective drugs to be soon available and prescribed with consequences of overtreatment, high costs, and poor outcomes [24]. Moreover, evidence from studies shows that drugs entering the market with a conditional market approval provided results later than expected, with a median delay of 274 days [25]. Among the fifteen (11%) drugs that received a conditional marketing authorization over the 6-year period, one (7%) was withdrawn from use in the European Union after four years from the approval date (2015). Indeed, olaratumab, originally designated as an orphan medicine, was approved in combination with doxorubicine with a conditional marketing authorization, on the basis of a randomized phase Ib–II that showed prolongation of PFS compared to doxorubicin alone. Later, in 2019, olaratumab was withdrawn from use in the European Union because of the lack of efficacy demonstrated by the phase III ANNOUNCE trial [9].

PRO end-points have been frequently incorporated in clinical trials (81%) but reported in the dossiers for EMA evaluation in a lower percentage (67%) of cases. They are the standard tool to report a patient’s subjective experience in prospective clinical studies. Historically, some methodological difficulties have limited the impact of PRO data on regulatory decisions (missing and low-quality data, bias, timing of assessment, single-dimensional PRO measures). Grossmann et al., analyzing cancer drugs approved by the EMA between 2009 and 2015, showed that 53% were lacking available data on QoL outcomes at the time of approval, and still one third of them (31%) after a monitoring period of three years [26]. EMA encourages the inclusions of PRO measures as outcomes in oncology clinical studies where they may provide added value for benefit risk balance [27]. They may be used as primary end-points in late line therapy studies, in patients with cancer-related symptoms, and in combination with other primary end-points (time to symptomatic tumor progression). Other examples of situations where PROs may be informative are the palliative setting and trials with best supporting care as control arm. PRO data may be more informative for confirmatory trials than the exploratory ones [2].

Some instruments have been introduced to strengthen the ability to identify the real benefit of anti-cancer therapy. To support treatment decision-making, the European Society for Medical Oncology (ESMO) developed a validated tool, the ESMO Magnitude of Clinical Benefit Scale (ESMO-MCBS), that calculates a relative ranking of the magnitude of clinical benefit of oncologic drugs, based on end-points, quality of life, and toxicity data [28]. Recently, Grössmann et al., applied the ESMO-MCBS tool to cancer drug indications approved over the last 12 years by EMA. They demonstrated that only a minority of indications (33%) met criteria for significant clinical benefit [29].

This study has some limitations. First, it did not explore secondary end-points and the post-marketing period; therefore, we are not able to provide evidence of efficacy for drugs that received a conditional marketing authorization. However, our aim was to offer a picture of EMA oncologic drugs approval at the time of their authorization. Second, our estimate of the absolute gain in survival was based on randomized trials where a comparison was available and may underestimate the real benefit of several signal transduction inhibitors that were often approved without a control arm.

More stringent rules should be required by regulatory agencies in post-marketing surveillance of oncologic drugs, both those who received conditional marketing authorization and those approved with surrogate end-points. This is crucial to estimate the real-world effectiveness. The ESMO-MCBS framework may be integrated in the assessment of drug dossiers and used for reimbursement negotiations, especially for drugs with low or questionable clinical benefit. The involvement of patients’ organizations and cancer patients in Regulatory Authorities evaluation is strongly encouraged.

## 5. Conclusions

In conclusion, our analysis confirms a long-standing trend in drug development for which most of the developmental efforts are concentrated on evaluating multiple drugs, sharing the same mechanism of action (“me-too drugs”), in the most frequent tumors, using surrogate endpoints. Also taking into account the strong unmet need for having novel, effective treatments for (even the most rare) tumors, we strongly believe that regulators should raise the bar for the approval of more clinically beneficial drugs and promote the technological advance.

## Figures and Tables

**Table 1 cancers-14-00889-t001:** Oncologic indications divided for mechanism of action.

Drug Classes	N Indications (%)
Signal trasduction inhibitor	42 (32)
Immunochemotherapy	40 (30)
Cell cycle and DNA repair mechanism	18 (14)
Angiogenesis inhibitor	16 (12)
Chemotherapy	8 (6)
Hormone therapy	7 (5)
Radiometabolics	1 (1)
Total	132

**Table 2 cancers-14-00889-t002:** Indications for solid tumors.

Cancer	N° Indications (%)
**Lung**	34 (26)
NSCLC	32
SCLC	2
**Breast**	21 (16)
**Genitourinary**	21 (16)
Renal cell	8
Prostate	7
Urothelial	6
**Skin**	16 (12)
Melanoma	13
Squamous/basocellular	2
Merkel	1
**Gastro-intestinal**	15 (11.2)
Colorectal	5
Hepatocellular	5
Pancreatic	2
Gastric	1
GIST	1
Esophageal	1
**Gynecological**	11 (8.3)
Ovarian	10
Cervix	1
**Head and neck**	4 (3)
**Thyroid**	4 (3)
**Soft tissue sarcoma**	2 (1.5)
**Neuroendocrine**	2 (1.5)
**NTRK positive**	2 (1.5)
Total	132 (100)

NSCLC: non-small cell lung cancer; SCLC: small cell lung cancer.

**Table 3 cancers-14-00889-t003:** Features of clinical trials leading to oncologic drug approvals.

Characteristic	Item	Number	Percentage (%)
**Phase**	I	3	2
	I–II	9	7
	II	21	16
	II–III	2	2
	III	97	73
**Hypothesis**	Superiority	100	76
	Non-inferiority	6	5
	Superiority and non-inferiority	2	1
	Exploratory/descriptive/NA	24	18
**Mean sample size** *(range)*	-	603.5 (12–4805)	-
**Randomization**	Yes	109	82
	No	23	18
**Control arm**	Active treatment with PL	24	18
	Active treatment without PL	62	47
	PL/BSC	23	17.5
	No control arm	23	17.5
**Experimental arm**	Monotherapy	77	58
	Combo	55	42
**Crossover**	Yes	15	11
	No	117	89
**Primary Endpoint**			
Advanced disease	OS	51	42
	PFS	42	35
	ORR	26	22
	Other outcomes	1	1
Non advanced	OS	1	8
	Other survival outcomes	9	75
	PK/pCR	2	17
**Quality of life**			
Evaluated by trial	Yes	107	81
	No	25	19
Reported in EPAR	Yes	88	67
	No	44	33
Descriptive	No difference between EX and CT arm	55	42
	In favor of EX arm	21	16
	In favor of CT arm	3	2
	Not applicable	53	40

PL: placebo; EX: experimental; CT: control.

## Data Availability

The data presented in this study are available in this article (and Appendix A).

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
