# Peer review of "Oncologic Drugs Approval in Europe for Solid Tumors: Overview of the Last 6 Years"

_cancers, 2022, doi:10.3390/cancers14040889_

Round 1

Reviewer 1 Report

Review of Manuscript ID cancers-1589830

Title: ‘Oncologic Drugs Approval in Europe for Solid Tumors: Over-2 view of The Last 6 Years’

Overview

The review article presents a very interesting account of the approvals of new therapeutic indications of existing drugs for solid tumors. The subject as such attracts high interest.  The paper gives extensive account of the cases with details of the clinical trials that approvals were based upon. The writing style however can be improved by avoiding linguistic improvisations and with adherence to scientific writing. Also, some short conclusions could be added after major sections of the Results part and a more extensive overall conclusion reflecting the main outcomes of the review is needed.

I suggest the following specific comments on grammar and readability that authors may want to take into consideration.

Abstract

Line 13

Change to ‘extension of indications’

Line 20 – remove ‘increasingly’

Introduction

Line 48 – Replace ‘To enter the market … patient’ with ‘To implement a new indication a positive benefit-risk …’…’

Line 61 – something is missing after have been

Line 62 -perhaps change ‘chance’ to ‘challenge’

Lines 67-70 – Text is not understood

Materials and methods

Line 123 – ‘  Data were extracted independently by three investigators’ is not clear what authors want to say. Individual contributions are added at the end of the main text.

Line 127 - What kind of test was performed? add more details

Results

Line 168 – The title should read ‘Features of Main Approved Studies

Line 173 - Please define 'latter scenario'

Discussion

Line 234 – Ref [10] should be after Apolone et al

Line 251 – The same with ref [12]

Line 337 – Please replace ‘of’ with ‘to’

Conclusions

The opinion of the authors for the correctness or hastiness of the approval process by the Health authorities with justification should be commented in the Conclusions section.

Author Response

We  want to strongly thank for the comment to our manuscript  that we acknowledge could be  improved.  We addressed all the suggestion and importantly the whole text benefitted in both the grammar/semantic ground and readibility.  Please find attached a point-by point review

Best Regards

Reviewer 2 Report

This is a very interesting and important review prepared by the group of  Genneralo Daniele from Italy presenting the recent drug development in oncology.

The Authors analyzed all new  indications (132) for 62 solid tumors issued by European Medicines Agency between 2015 and 2020. In about half of indications, the approval was biomarker-based. Immune check point inhibitors and signal transduction inhibitors were the two most representative drug categories. Most of the indications were for the advanced setting and front-line therapy. The most common tumor types were non-small cell lung cancer, breast, and melanoma. Two thirds of the indications were approved based on phase III trials. However, 12 indications were based only on early phase (I or I/II) trials. As emphasized by the Authors, the majority of drugs have entered the market without evidence of overall survival or quality of life benefit but based on surrogate outcomes. Along this line, many drugs with the same, or a very similar mechanism of action,were approved for the same indication without a direct head-to head comparison. These data point to an urgent need to develop optimal methodology for assessing effectiveness and safety of new anticancer drugs.

Author Response

We want to thank the reviewer for the comment on our manuscripts. We are very pleased that the message was caught by this valuable expert.

Best Wishes